# Preparation of High Specific Surface Area Activated Carbon from Petroleum Coke by KOH Activation in a Rotary Kiln

**Kechao Wang** and **Shaoping Xu** *

School of Chemical Engineering, Dalian University of Technology, No. 2 Linggong Road, Dalian 116024, China; kechaowang@outlook.com
* Correspondence: spxu@dlut.edu.cn; Tel.: +86-411-84986143

**Abstract:** In the preparation of high specific surface area activated carbon (AC) by KOH activation, the swelling of the reactant mixture and the particles' agglomeration deteriorates the process and the property of product. In this study, a novel method using a rotary kiln loaded with steel balls has been developed for the preparation of AC from petroleum coke (PC) by KOH activation. It has been found that the molten KOH caused the swelling of the reaction mixture at a lower activation temperature, while the molten $K_2O$ led to the particles' agglomeration at a higher temperature. The steel balls could relieve the swelling and agglomeration and enhance the pore structure development of the AC by boosting the heat and mass transfer in the reactor. At an activation temperature of 800 °C and a KOH/PC mass ratio of 3:1, the specific surface area of the AC obtained without the addition of steel balls in the kiln is 1492 $m^2/g$, while that with the steel balls is 1996 $m^2/g$. The introduction of $CO_2$ during the activation could further decrease the particles' agglomeration by converting the $K_2O$ into thermoset $K_2CO_3$ and develop more mesopores of AC. Specifically, the average pore size of the AC increased from 2.20 to 2.72 nm.

**Keywords:** activated carbon; KOH; activation; rotary kiln; petroleum coke; high specific surface area





## 1. Introduction

High specific surface area activated carbon (AC) is widely used as an adsorbent for gas storage, a catalyst support and an electrode of electrochemical capacitors because of its high surface area and well-developed microporous structure [1–4]. It is generally produced from raw materials with high carbon content, such as petroleum coke (PC) and anthracite, by alkaline hydroxides such as NaOH and KOH activation. In most cases, KOH is chosen as the activation agent because it exhibits superior activation activity than NaOH [5], yielding an AC with a higher specific surface area and a more abundant pore structure. In this process, a large amount of KOH, i.e., a high ratio of KOH to the carbonaceous feedstock, has to be used, which leads to high production costs, reactor corrosion and safety risks [6–8]. Specifically, the activation process involves melting of the thermoplastic reactants, formation of a viscous paste of the reaction mixtures and its swelling and adhesion to the reactor wall [9,10], which seriously deteriorate the production efficiency and product quality.

In addition to KOH with a melting point of 360 °C [11], other thermoplastic K-containing compounds may form during the activation, which varies with the specific activation process. For instance, based on the activation of 3,5-dimethylphenol formaldehyde resin with NaOH [12] and the activation of coal with KOH [13], a reaction mechanism as shown by Equations (1) and (2) was proposed. It is indicated that KOH would attack the organic structure of the feedstock to form $K_2CO_3$, $K_2O$ and $H_2$ at 400–600 °C. The generated $K_2CO_3$ and $K_2O$ could further react with carbon (C) to form metallic K and CO at temperatures higher than 600 °C (Equations (3) and (4)).

$$4KOH + \text{\textbackslash}CH_2 \rightarrow K_2CO_3 + K_2O + 3H_2 \tag{1}$$

$$8KOH + 2\text{\textbackslash}CH \rightarrow 2K_2CO_3 + 2K_2O + 5H_2 \tag{2}$$

$$K_2O + C \rightarrow 2K + CO \tag{3}$$

$$K_2CO_3 + 2C \rightarrow 2K + 3CO \tag{4}$$

Lillo-Rodenas et al. [14,15] calculated the standard Gibbs free energy of possible reactions during the KOH activation of Spanish anthracite and proposed that KOH could react with C to form K, $H_2$ and $K_2CO_3$ at 730 °C (Equation (5)).

$$6KOH + 2C \rightarrow 2K + 3H_2 + 2K_2CO_3 \tag{5}$$

Our previous studies [16,17] showed that, in the low temperature region (<600 °C), KOH could react with the "active carbons", such as $\text{\textbackslash}CH$ and $\text{\textbackslash}CH_2$, in the feedstock PC to form $K_2CO_3$ and $K_2O$. At higher temperatures up to 830 °C, the $K_2CO_3$ and $K_2O$ could further react with $\text{\textbackslash}CH$ and $\text{\textbackslash}CH_2$ rather than C in PC to yield K. The reactions are shown as follows.

$$K_2CO_3 + 2\text{\textbackslash}CH \rightarrow 2K + 3CO + H_2 \tag{6}$$

$$K_2CO_3 + \text{\textbackslash}CH_2 \rightarrow K_2O + 2CO + H_2 \tag{7}$$

$$K_2O + \text{\textbackslash}CH_2 \rightarrow 2K + CO + H_2 \tag{8}$$

$$2K_2O + 2\text{\textbackslash}CH \rightarrow 4K + 2CO + H_2 \tag{9}$$

Among the above-mentioned K-containing compounds, $K_2O$ has a melting point of 740 °C [18], which is just within the ordinary activation temperature range. As enough $K_2O$ presents, it might also cause the reaction materials to form a viscous paste or even agglomerate into lumps. After the activation, the residual thermoplastic K-containing compounds, in the form of either viscous paste or the particles' agglomerates, would solidify as the temperature decreases. The hard lumps formed on the reactor wall would be difficult to peel off.

Even though the KOH activation mechanisms have been well studied, the effects of the viscous paste and the particles' agglomeration on the activation have long been overlooked in laboratory studies. The reason might be that most of the activation experiments in laboratories have been carried out intermittently in boat or tank reactors with a carbonaceous feedstock of usually no more than 5 g [19–22]. In these cases, the aforementioned problems are generally not serious.

In the massive production of AC, however, the swelling and agglomeration of the reactant mixture may damage the activation and impede its normal operation, especially in the case of continuous production. To alleviate the swelling and agglomeration problem, some measures were applied in the production of AC. For example, Standard Oil Company (Indiana) in Chicago, IL, USA used two indirectly fired rotary tube calciners in series to produce AC [23]. The mixture of carbonaceous feedstock and KOH was pre-activated at a lower temperature, around 500 °C, in the first calciner equipped with a rotating auger. The agitation by the auger would favor the discharge of the gases and steam produced, thereby reducing the swelling. The product of the first calciner was then transferred to the second calciner without a rotation auger to carry out the activation around 800 °C. Here, the agitation was provided only by the rotation of the calciner, and the particles' agglomeration at the higher temperature was not taken into account. Similarly, Kansai Netsukagaku Kabushiki Kaisha in Amagasaki, Japan produced AC also in two temperature stages but in a single vertical calciner with a stirrer [24–26]. Both of the aforementioned technologies tried to relieve the swelling or agglomeration problem by mechanical stirring.

Nevertheless, the effects of both the rotating auger and the stirrer on these problems have not yet been revealed.

In this study, to elucidate the impact of the swelling and agglomeration on the activation and development of an efficient AC production technology, preparation of AC from PC by KOH activation in a rotary kiln has been carried out. Specifically, the kiln has a processing capacity of up to 15 g PC plus 3 times KOH. The formation of non-condensable gases, $H_2O$ and K-containing compounds (K, $K_2CO_3$ and $K_2O$) with the reaction temperature and time on stream has been investigated. To relieve the swelling and agglomeration problem, both mechanical and chemical measures, i.e., loading steel balls and introducing $CO_2$ into the rotary kiln, respectively, have been adopted. The effects of both measures on the pore development of AC have been studied.

## 2. Materials and Methods

### 2.1. Raw Material

PC ground to less than 150 μm from Liaoyang Petrochemical Company in Liaoyang, China and KOH (analytical reagent) from Tianjin Kermel Chemical Reagent Co., Ltd. in Tianjin, China were used as feedstocks to prepare the AC. The results of the PC's proximate analysis based on an air-dry basis (ad.) and an ultimate analysis on the dry and ash-free basis (daf.) are listed in Table 1.

**Table 1.** Proximate and ultimate analysis data of PC.

| Proximate Analysis (wt.%, ad.) | | | | Ultimate Analysis (wt.%, daf.) | | | | |
|---|---|---|---|---|---|---|---|---|
| **Ash** | **Moisture** | **Volatile** | **Fixed Carbon** | **C** | **H** | **O \*** | **N** | **S** |
| 0.28 | 0.83 | 8.80 | 90.09 | 91.16 | 3.59 | 2.40 | 2.59 | 0.26 |

\* by difference.

### 2.2. Preparation of AC

The preparation process of AC is shown in Figure 1. Approximately 15 g PC were mixed with KOH of different KOH/PC mass ratios at room temperature. The mixture was then put into the rotary kiln filled with steel balls, as shown in Figure 2. The rotary kiln has a cavity 100 mm in length and 130 mm in diameter. Its inlet and outlet pipes both have a diameter of 45 mm. The kiln-loaded steel balls consist of 17 pieces 20 mm in diameter and 20 pieces 14 mm in diameter. Under a $N_2$ flow of 90 mL/min, the mixture in the kiln was electrically heated at a rotating speed of 60 r/min and a heating rate of 10 °C/min from room temperature to the activation temperature and was held at this temperature for a period of time. When the reaction was over, the reactor was cooled down to room temperature under $N_2$. A part of the solid product defined as the unwashed sample was used to evaluate the particles' agglomeration and quantify the K-containing compounds in the solid products. The remaining solid product was washed with hydrochloric acid of 1 mol/L and distilled water successively until the filtrate became pH neutral. The solid was then dried to obtain AC.

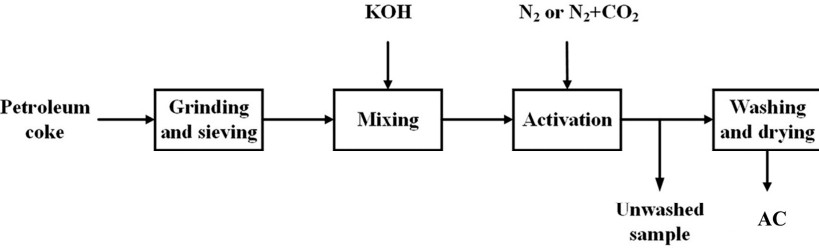

**Figure 1.** Preparation process of AC.

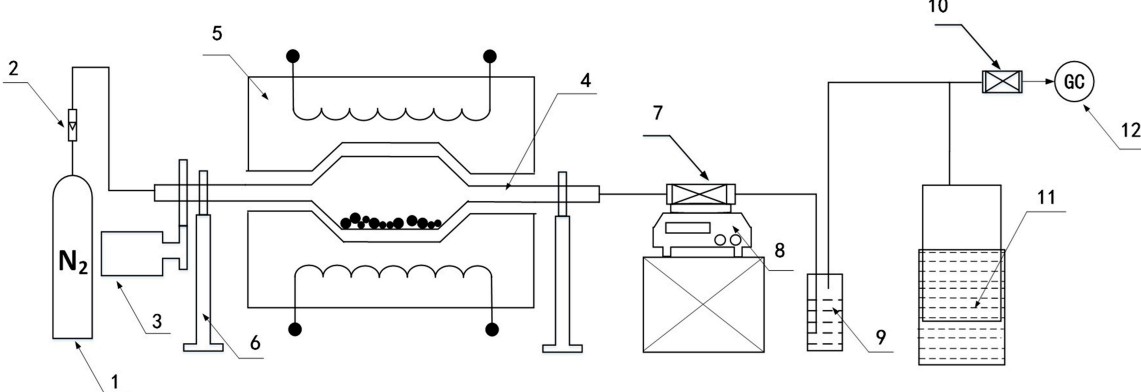

**Figure 2.** Schematic diagram of rotary kiln. 1. High purity nitrogen; 2. Mass flow controller; 3. Motor and driving gear; 4. Rotary kiln loaded with steel balls; 5. Electric furnace; 6. Supporting bearing; 7. Absorber; 8. Electric balance; 9. Gas purification bottle; 10. Dryer; 11. Gas cabinet; 12. Gas chromatography.

*2.3. Analysis of Activation Products*

2.3.1. Analysis of Gas, Water and K-Containing Compounds

The non-condensable gases were analyzed by a GC7900 gas chromatography (GC) equipped with a thermal conductivity detector (TCD) and a flame ionization detector (FID). The total gas volume was determined by a calibrated gas cabinet.

The steam entrained in the product gas was absorbed by the $CaCl_2$ filled in the absorber at the outlet of the rotary kiln. The water mass was measured by the weight increase of the absorber.

The K remained after the activation was quantified as follows: cooling down the rotary kiln to 150 °C and then introducing distilled water into it. The K reacted with the water in stoichiometry, as shown in Equation (10), to form $H_2$.

$$2K + 2H_2O \rightarrow 2KOH + H_2 \tag{10}$$

The $H_2$ generated was collected in the gas cabinet and analyzed by the GC. The amount of K is calculated by:

$$n(K) = 2 \times c(H_2) \times V \tag{11}$$

where $n(K)$ is the amount of K in mol; $c(H_2)$ is the $H_2$ concentration in the collected gas in mol/L; and V is the total volume of the collected gas in L.

The $K_2CO_3$ remained after the activation was measured based on a precipitation reaction. The solid activation products were washed with distilled water until all the K-containing compounds were dissolved in the water. $BaCl_2$ was then added into the water solution to transform the $K_2CO_3$ in it into $BaCO_3$. As a precipitate, the $BaCO_3$ was subsequently filtered and dried. The amount of $K_2CO_3$ is calculated by:

$$n(K_2CO_3) = m(BaCO_3)/M(BaCO_3) \tag{12}$$

where $n(K_2CO_3)$ is the amount of $K_2CO_3$ in mol; $m(BaCO_3)$ is the mass of $BaCO_3$ in g; and $M(BaCO_3)$ is the relative molecular mass of $BaCO_3$ in g/mol.

After the activation, the $K_2O$ was analyzed by titrating the filtrate from the filtration with hydrochloric acid (HCl), as mentioned above. The amount of $K_2O$ in the activation products is calculated by:

$$n(K_2O) = 0.5 \times V(HCl) \times c(HCl) \tag{13}$$

where $n(K_2O)$ is the amount of $K_2O$ in mol; $V(HCl)$ is the volume of HCl in L used in the titration; and $c(HCl)$ is the concentration of HCl in mol/L.

### 2.3.2. Characterizations of AC

The porous texture parameters of AC were obtained based on the adsorption isotherms of $N_2$ at 77 K measured by a BWJK-122W adsorption apparatus. Before the test, the samples were subjected to vacuum degassing at 250 °C for 2.5 h. The specific surface areas were calculated with the Brunauer–Emmett–Teller (BET) equation. The micropore volumes were calculated by constructing t-plots. The total pore volumes were estimated from the liquid volume of nitrogen adsorbed at relative pressure $p/p_0 = 0.98$.

An XRD analysis of the AC samples was conducted using a Shimadzu XD-3A X-ray diffractometer made by Shimadzu Corporation in Kyoto, Japan. The X-ray wavelength was $\lambda = 0.15405$ nm, and the scanning range was 10° to 80° with a step length of 0.02°/s.

### 2.3.3. Evaluation of Particles' Agglomeration

The particle size distributions of the solid products were used to evaluate the particles' agglomeration. After activation, the unwashed samples were sieved in a vibrating screen for 10 min to obtain agglomerates of the following particle size groups: <0.4 mm, 0.4–0.9 mm, 0.9–2 mm, 2–5 mm and >5 mm. The agglomerates were formed from the carbonaceous particles' agglomeration by the residual thermoplastic K-containing compounds, such as KOH and $K_2O$. The higher the ratio of the larger agglomerates, the more severe the particles' agglomeration.

## 3. Results and Discussion

### 3.1. Products Evolution during Activation

A mixture of 15 g PC and 45 g KOH was adopted as feedstock to explore the evolution of various products during the activation of PC. The activation was carried out by heating the mixtures from room temperature to the final temperature at a rate of 10 °C/min and holding at this temperature up to 40 min.

### 3.1.1. Non-Condensable Gases

The evolution rates (in mL/min per gram of PC) of the non-condensable gases' components, namely, $H_2$, CO, $CH_4$ and $CO_2$, during the activation are shown in Figure 3.

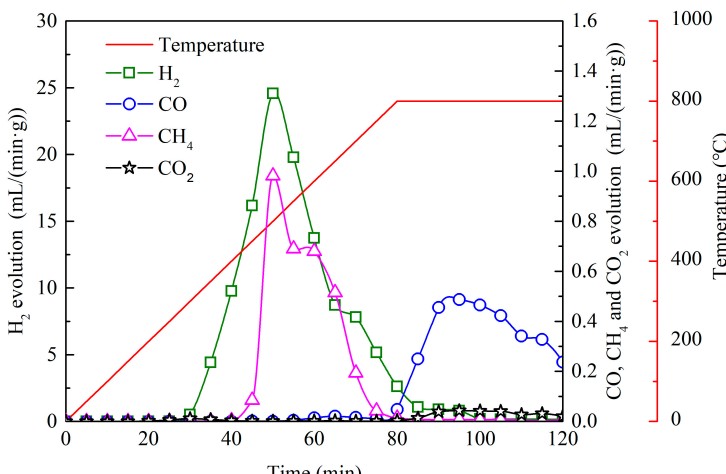

**Figure 3.** The evolution of non-condensable gases during the activation.

It could be found from Figure 3 that the evolved gas consisted of mainly $H_2$. The majority of $H_2$ evolution began from 300 °C, reached the maximum at 500 °C and then decreased gradually to a low level at the final activation temperature 800 °C. The $H_2$ was produced by the reactions of KOH with PC as shown by the Equations (1), (2) and (5) [11,14].

The potassium compounds formed from these reactions, i.e., K, $K_2O$ and $K_2CO_3$, could take part in further reactions to release $H_2$, as indicated by the small $H_2$ evolution peak at around 700 °C. For example, K could react with the $H_2O$ generated from the decomposition of KOH (Equation (14)) [25,27].

$$2KOH \rightarrow K_2O + H_2O \tag{14}$$

The reactions between $K_2O$ or $K_2CO_3$ and PC (Equations (6)–(9)) would take place to yield $H_2$ and CO [16]. The water–gas reaction (Equation (15)) might also be a source of $H_2$ at temperatures over 700 °C.

$$C + H_2O \rightarrow CO + H_2 \tag{15}$$

The CO and $H_2O$ formed during the activation could further produce $H_2$ by a water–gas shift reaction (Equation (16)).

$$CO + H_2O \rightleftharpoons CO_2 + H_2 \tag{16}$$

Both reactions 15 and 16 would be promoted by the catalytic K-containing intermediates, such as K, $K_2CO_3$ and $K_2O$ [28,29]. Nevertheless, little CO and $CO_2$ were found below 800 °C because the $CO_2$ generated had almost all been absorbed by $K_2O$ to form $K_2CO_3$, and the shift rection could be moved forward, yielding reduced CO.

Apart from $H_2$, there were also two small $CH_4$ evolution peaks at around 500 °C and 600 °C, respectively. The former came from the PC pyrolysis, while the latter could be attributed to the methanation of CO (Equation (17)) [30].

$$CO + 3H_2 \rightleftharpoons CH_4 + H_2O \tag{17}$$

This reversible reaction could also be accelerated by the catalytic K-containing intermediates. Thermodynamically, the $CH_4$ and CO formation are favorable at the lower and the higher temperature, respectively. As a result, a relatively larger amount of $CH_4$ but little CO were detected at temperatures lower than 800 °C, while it was the reverse at 800 °C. The small amount of $CO_2$ evolution at the final temperature of 800 °C might come from the slight decomposition of $K_2CO_3$ [31].

### 3.1.2. Water

The amount of water produced by heating the mixture of PC and KOH is shown in Figure 4. For comparison, the water evolutions of both 45 g KOH and 15 g PC under the same experimental condition are also displayed.

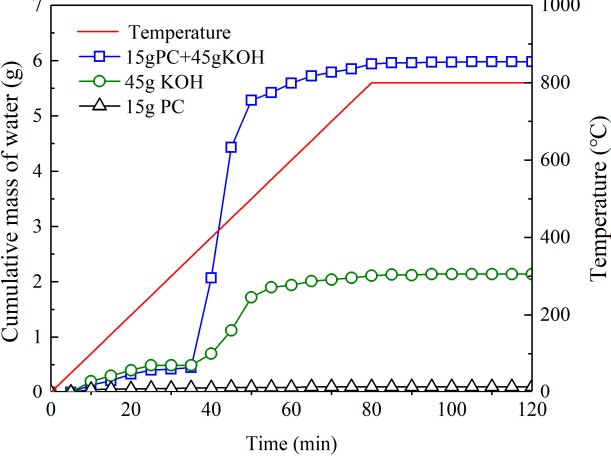

**Figure 4.** Water evolution by heating PC, KOH and the mixture of PC and KOH.

As Figure 4 shows, the water evolution of the mixture of PC and KOH could be characterized into three stages with the temperature increase. Before 350 °C, only a small amount of water was produced; during 350–500 °C, the majority of the water was generated; and from 500 °C up, the water evolution decreased rapidly and finished at the final temperature of 800 °C. As a comparison, the PC alone yielded almost no water during the entire heating process. The water evolution of KOH behaved almost the same as the mixture of PC and KOH, but its quantity was only about 1/3 of that of the mixture of PC and KOH. At the higher temperature, no significant water evolution was found.

It is suggested that the water evolution of the mixture of PC and KOH below 350 °C was from the evaporation of the water adsorbed by the alkali [32]. In the temperature range of 350–600 °C, a small part of the water was yielded from the dehydration of KOH (Equation (14)) [23,25]. The major water evolution above 350 °C came from the chemical interactions between KOH and PC (Equations (18) and (19) [33] and Equations (20) and (21)). In addition, the methanation of CO (Equation (17)) also contributed to the water evolution [17].

$$4KOH + C \rightarrow 4K + CO_2 + 2H_2O \tag{18}$$

$$2KOH + CO_2 \rightarrow K_2CO_3 + H_2O \tag{19}$$

$$6KOH + {\rangle}CH_2 \rightarrow 4K + K_2CO_3 + 3H_2O + H_2 \tag{20}$$

$$12KOH + 2{\rangle}CH \rightarrow 8K + 2K_2CO_3 + 6H_2O + H_2 \tag{21}$$

### 3.1.3. K and K-Containing Compounds

The XRD patterns of the unwashed samples of varied activation temperatures from the PC and KOH mixture are given in Figure 5. The soaking time at the final activation temperatures was the same of 10 min. As it is shown, K, $K_2O$ and $K_2CO_3$ could be found in all the samples, while no KOH was detected when the activation temperature was over 600 °C, indicating that KOH had been completely converted into K, $K_2O$ and $K_2CO_3$.

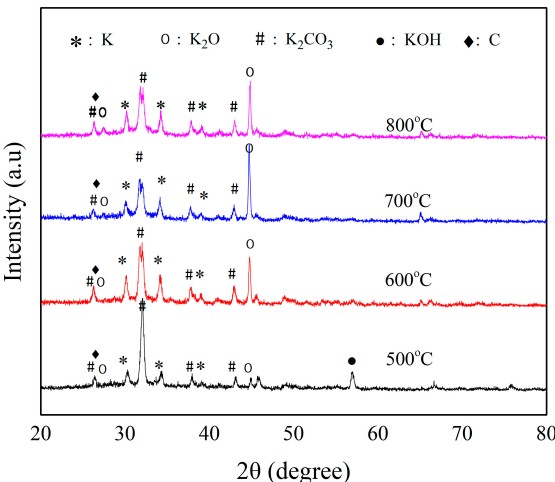

**Figure 5.** XRD patterns of unwashed activated samples at different temperatures.

The K, $K_2O$ and $K_2CO_3$ in the samples were quantified in mmol per gram PC based on the methods described in Section 2.3.1, and the results are shown in Figure 6. Because KOH and $K_2O$ coexisted at temperatures lower than 600 °C, as shown in both Figures 4 and 5, and they could not be differentiated by the acid–base titration, here, only the $K_2O$ generated over 600 °C is presented.

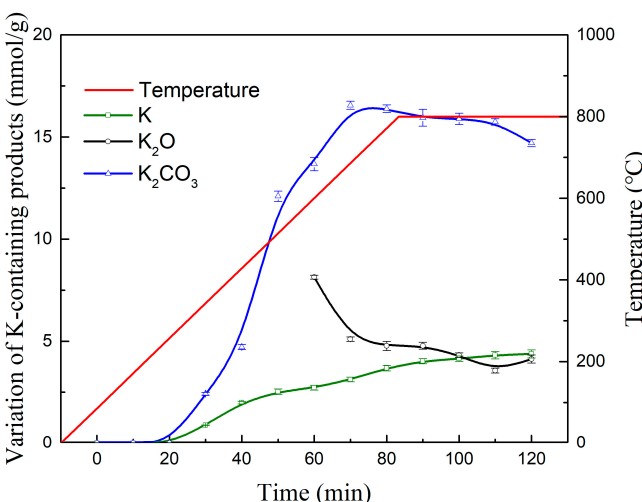

**Figure 6.** K, $K_2CO_3$ and $K_2O$ in the activation process.

With the progress of activation, the amount of K increased steadily. At temperatures lower than 600 °C, the K came from the reactions of KOH with the PC (Equations (5), (18), (20) and (21)). When the temperature was over 600 °C, the reactions of $K_2CO_3$ and $K_2O$ with the PC (Equations (6), (8) and (9)) could be the source of K.

The $K_2O$ decreased gradually with the temperature increase. A part of the $K_2O$ was transformed into K and $K_2CO_3$, as indicated by Equations (8) and (9) [14,34]. The carbonation of $K_2O$ by $CO_2$ generated during the activation also led to a decrease of $K_2O$.

The $K_2CO_3$ increased rapidly below 700 °C and then decreased slowly over time at the final temperature 800 °C. The increase of $K_2CO_3$ came mainly from the reactions between KOH and the PC (Equations (1), (2), (5) and (19)–(21)). The carbonation of $K_2O$ might also contribute to the increase of $K_2CO_3$, as it was closely related to the decrease of $K_2O$ from 600 °C to 700 °C. The decrease of $K_2CO_3$ could be attributed to its reactions with the PC (Equations (6) and (7)) [16].

### 3.2. Measures to Reduce the Swelling and Agglomeration

#### 3.2.1. Mechanical Measure

As discussed above, the thermoplastic KOH remained until 600 °C. At temperatures over 360 °C, KOH would melt, and the relatively larger amount of KOH would combine the carbonaceous particles to form a viscous paste. In the temperature range between 360 °C and 600 °C, a large amount of gas was produced upon the activation. Its evolution from the viscous paste would cause swelling of the reaction mixture. Furthermore, the majority of $H_2O$ was generated in this temperature range, which would aggravate the swelling by converting $K_2O$ and K formed during the activation into KOH again. Therefore, it is reasonable to disperse the sticky bulk by mechanical stirring and to separate the activation into two activation temperature stages to alleviate the swelling caused by KOH and $H_2O$, as both Standard Oil Company (Indiana) [23] and Kansai Netsukagaku Kabushiki Kaisha in Amagasaki, Japan [24–26] demonstrated.

To mechanically disperse the thermoplastic components in the reaction mixture and to assist the product gas release, in this study, a rotary kiln filled with steel balls was applied. The effect of the steel balls on the activation was investigated by comparing the processes without and with the steel balls under the same activation condition, i.e., the final temperature of 800 °C and the soaking time of 40 min at this temperature.

The pore structure parameters of the AC prepared with different KOH/PC ratios in the cases without and with the steel balls are shown in Table 2. As the KOH/PC ratio rose, the specific surface area ($S_{BET}$) of AC increased, the average pore size ($D_a$) and mesopore volume ($V_{meso}$) decreased, and the micropore volume ($V_{micro}$) and total pore volume ($V_t$) increased in both cases. At the same KOH/PC ratio, the $S_{BET}$, $D_a$, $V_{micro}$ and $V_t$ of AC

with the balls loaded were larger than those without the steel balls. It is indicated that the presence of the steel balls could enhance the pore structure development of AC. The reason is that the steel balls' movement improved the heat and mass transfer of the activation by dispersing the thermoplastic components, promoting the gas discharging, reducing the swelling of the viscous paste and enhancing the contact between the activation agents and the carbonaceous feedstock.

**Table 2.** Pore structure parameters of the AC in the cases without and with steel balls.

| Rotary Kiln State | KOH/PC | $S_{BET}$ (m$^2$/g) | $D_a$ (nm) | $V_t$ (cm$^3$/g) | $V_{micro}$ (cm$^3$/g) | $V_{meso}$ (cm$^3$/g) |
|---|---|---|---|---|---|---|
| Without steel balls | 1:1 | 538 | 2.82 | 0.38 | 0.22 | 0.16 |
| | 2:1 | 1179 | 2.27 | 0.67 | 0.40 | 0.17 |
| | 3:1 | 1492 | 2.01 | 0.75 | 0.67 | 0.09 |
| With steel balls | 1:1 | 1114 | 3.30 | 0.92 | 0.48 | 0.44 |
| | 2:1 | 1681 | 2.37 | 1.00 | 0.72 | 0.28 |
| | 3:1 | 1996 | 2.20 | 1.10 | 0.82 | 0.28 |

The loading of steel balls into the rotary kiln could also decrease the particles' agglomeration. As shown in Figure 7, in the case without loading the steel balls, there was a large number of bigger agglomerates in the activation residue, and their proportion increased rapidly with the increase of the KOH/PC ratio. Upon adding the balls, however, the bigger agglomerates decreased while the smaller agglomerates increased, evidently under all KOH/PC ratios. It is suggested that the movements of the steel balls, especially friction and collision, could impact and alleviate the particles' agglomeration to some extent during the solidifying of the thermoplastic components.

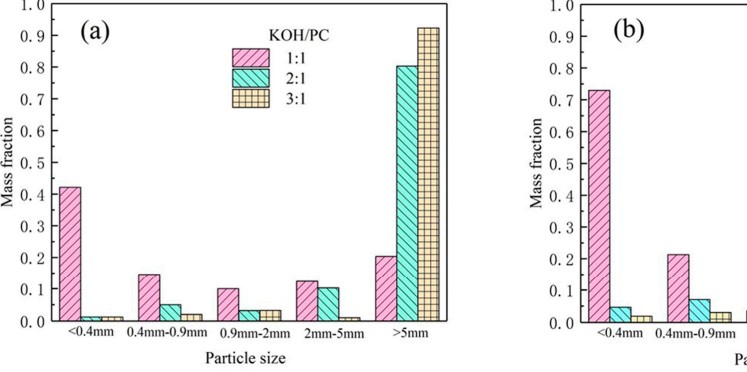

**Figure 7.** Particle size distributions of the unwashed samples in the cases without (**a**) and with (**b**) steel balls.

### 3.2.2. Chemical Measure

As mentioned above, $K_2O$ was the major thermoplastic component in the activation residues. It might not cause severe swelling at the temperature over its melting point of 740 °C because its quantity was not too high and the gas evolution at this temperature was relatively small. Nevertheless, it would lead to the particles' agglomeration as the temperature decreased after the activation. Therefore, in addition to the mechanical measure, a chemical measure, i.e., converting the $K_2O$ into a thermoset substance, has been used to further reduce the agglomeration. Specifically, as a carbonation reagent to transform $K_2O$ into $K_2CO_3$, $CO_2$ was introduced into the kiln at 90 mL/min with steel balls at the final activation temperature 800 °C for 40 min.

Figure 8 shows the particle size distributions of the unwashed samples in the case without and with $CO_2$ introduction. After the introduction of $CO_2$, the proportion of the smaller agglomerates had significantly increased, especially at a higher KOH/PC ratio. It could be concluded that the introduction of $CO_2$ effectively relieved the particles'

agglomeration. The reason is that $CO_2$ could convert the thermoplastic $K_2O$ to thermoset $K_2CO_3$, which prevented the particles' agglomeration in the process of the temperature decreasing after the activation.

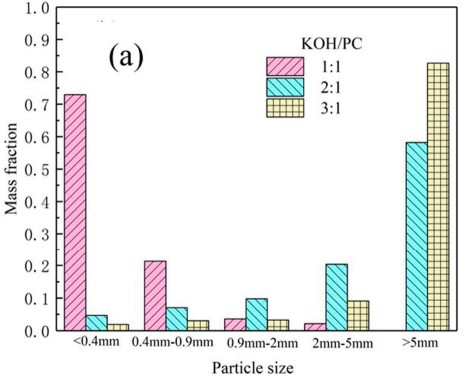 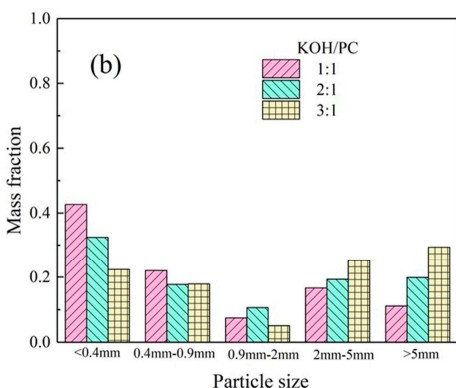

**Figure 8.** Particle size distributions of the unwashed samples in the cases without (**a**) and with (**b**) $CO_2$ introduction.

The pore structure parameters of the ACs prepared with different KOH/PC ratios in the cases without and with the introduction of $CO_2$ are shown in Table 3. Compared with the sample without $CO_2$, the sample with $CO_2$ introduction has smaller $S_{BET}$ and $V_{micro}$ but larger $D_a$ under the same KOH/PC ratio. At the KOH/PC mass ratio 3:1, the introduction of $CO_2$ led to an increase in the average pore size of AC from 2.20 to 2.72 nm and the $V_{meso}$ from 0.28 $cm^3/g$ to 0.45 $cm^3/g$, and the $S_{BET}$ was maintained at 1789 $m^2/g$. It is indicated that the contribution of $K_2CO_3$ to the activation process was enhanced especially under the higher KOH/PC ratio, which promoted preferentially the formation of mesopores in the AC [35]. In addition, the reaction of $CO_2$ with the PC in presence of the catalytic K-containing compounds, i.e., the Boudouard reaction, could also take place at this temperature to develop the pore structure of the AC [36].

**Table 3.** Pore structure parameters of AC in the cases without and with $CO_2$ introduction.

|  | KOH/PC | $S_{BET}$ (m$^2$/g) | $D_a$ (nm) | $V_t$ (cm$^3$/g) | $V_{micro}$ (cm$^3$/g) | $V_{meso}$ (cm$^3$/g) |
|---|---|---|---|---|---|---|
| Without $CO_2$ | 1:1 | 1114 | 3.30 | 0.92 | 0.48 | 0.44 |
|  | 2:1 | 1681 | 2.37 | 1.00 | 0.72 | 0.28 |
|  | 3:1 | 1996 | 2.20 | 1.10 | 0.82 | 0.28 |
| With $CO_2$ | 1:1 | 891 | 3.50 | 0.78 | 0.38 | 0.40 |
|  | 2:1 | 1483 | 2.88 | 1.07 | 0.65 | 0.42 |
|  | 3:1 | 1789 | 2.72 | 1.22 | 0.77 | 0.45 |

## 4. Conclusions

A new preparation method of AC from PC by KOH activation using a rotary kiln loaded with steel balls has been developed. The mechanism and methods for relieving the swelling and agglomeration during the preparation of AC have been researched.

The evolutions of non-condensable gases, water and potassium compounds during the activation process were investigated to clarify the source of the swelling and agglomeration at different temperatures. It is shown that the molten KOH caused the swelling at activation temperatures lower than 600 °C, while the $K_2O$ led to the particles' agglomeration as the temperature decreased after the activation.

Both mechanical and chemical methods were applied to relieve the swelling and agglomeration and they were proved to be efficient. As the mechanical measure, the steel balls in the rotary kiln could not only enhance the pore structure development of AC by boosting the heat and mass transfer in the reactor, but also decrease the particles' agglomeration. At the KOH/PC ratio of 3:1, the specific surface area of AC reached

1996 $m^2/g$ after introducing the steel balls into the kiln, much higher than the value of 1492 $m^2/g$ without adding the steel ball. As the chemical measure, the introduction of $CO_2$ at the high-temperature stage was beneficial to the reduction in the particles' agglomeration by converting the thermoplastic $K_2O$ into the thermoset $K_2CO_3$. It also promoted the development of the mesopores of AC, which could be used in those special applications where a developed mesoporous structure is needed. At the KOH/PC mass ratio of 3:1, the introduction of $CO_2$ led to an increase in the average pore size of AC from 2.20 to 2.72 nm and the $V_{meso}$ from 0.28 $cm^3/g$ to 0.45 $cm^3/g$, while the specific surface area remained at 1789 $m^2/g$.

**Author Contributions:** Conceptualization, K.W.; data curation, K.W.; formal analysis, K.W.; methodology, K.W.; project administration, S.X.; resources, S.X.; writing–original draft, K.W.; writing–review and editing, S.X. All authors have read and agreed to the published version of the manuscript.

**Funding:** This research was funded by the National Natural Science Foundation of China (No. 21376046).

**Data Availability Statement:** Data are contained within the article.

**Conflicts of Interest:** The authors declare no conflicts of interest.

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
