# Peer review of "Preparation of High Specific Surface Area Activated Carbon from Petroleum Coke by KOH Activation in a Rotary Kiln"

_processes, doi:10.3390/pr12020241_

Round 1

Reviewer 1 Report

Comments and Suggestions for Authors

The article is very interesting and makes a good impression. The scientific novelty is average, since activation processes using alkalis are described in sufficient detail (as can be seen even from the list of references). Practical interest in the production of new highly efficient adsorbents makes the work relevant. Despite this,

The most serious remark requiring mandatory correction:

The X-ray diffraction data in Fig. 5 raises a question. Where is the carbon component? If the signature clearly states that this is washed activated carbon. According to X-ray diffraction pattern 5, this is not coal, but simply products of thermal transformation of sodium hydroxide (KOH)  in the presence of carbon

There are a number of relatively minor comments on the text of the work:

1.      For Table 1 it is necessary to provide a decoding of what the letters A, M, V, FC  mean

2.      There is also no explanation given why potassium hydroxide (NaOH) was used instead of the relatively cheap sodium hydroxide (KOH).

3.      After reading the article, a natural question arises: what is the practical application of the results? Why didn't the researchers conduct a rapid test of the adsorption of a typical dye «methylene blue» on coal samples? What is the cost of the activation process?

4.      Also of interest are practical environmental aspects: where should waste gases be directed? What to do with rinse water? what kind of waste will there be? It is necessary to give at least 1-2 sentences with explanations.

Comments on the Quality of English Language

good

Reviewer 2 Report

Comments and Suggestions for Authors

The authors of the paper presented a new approach to producing carbon with a highly developed surface. The proposed method is quite original. Despite this, the reviewer had some comments on the presented material.

1. In work [11], the interaction of sodium hydroxide with rubber was considered. Reactions (1) and (2) presented are incorrect. The authors [11] presented a different interaction mechanism. In addition, from the point of view of the chemistry of high-molecular compounds, reactions primarily occur with the functional groups of substituents and not the main chain. Moreover, the action of both acids and alkalis can lead to the cleavage of the double bond with the formation of organic radicals rather than the complete disappearance of the double bond. Reaction (2) is presented incorrectly!

2. It is necessary to provide a decoding of the symbols presented in Table 1. What do ad. and daf. mean? What is proximate and what is ultimate? What is FS, M, A, V?

3. For what reason were molecular nitrogen atmospheres and mixtures of molecular nitrogen and carbon dioxide chosen?

4. While reading the manuscript, the reviewer was unable to understand whether the presence of steel balls does not affect gas release in the system? Or do the processes presented by the authors refer to a system without balls?

5. Reactions (20) and (21) are also presented incorrectly. Authors should present the proposed mechanisms more accurately. More generally, the polymeric material is represented as part of a polymer repeat chain with the main functional group in square brackets. And once again, take a closer look at the primary sources.

6. Also, from the data presented, it is not clear that steel balls were added only when using a nitrogen atmosphere? Was the atmosphere of a mixture of nitrogen and carbon dioxide used without steel balls? This issue should be clarified.

7. I would also like to know the authors’ opinion on the question: is it preferable to obtain carbon with a more developed surface or with a larger pore size?

The above comments do not reduce the level of the article, but will allow us to clarify controversial and unclear points.

Reviewer 3 Report

Comments and Suggestions for Authors

This article discusses the preparation of high specific surface area activated carbon from petroleum coke in a rotary kiln. The paper is interesting and could be highly beneficial for researchers working in this field. It is well-structured and easily understandable. However, I believe there are some improvements that the authors should address before publication:

1.- Some sentences are rudimentarily constructed and could be clearer with a thorough text review.

2.- Equations 1 to 9 and 18 to 21 need to be better prepared.

3.- Certain units (e.g., m2·g-1) should be rewritten without the dot.

4.- Figure 5 needs improvement: the diffractograms could be enlarged slightly, and the line appearing below the figure should be removed.

5.- Figure 6 should include error bars for each measurement.

6.- Lastly, it would have been interesting to include an electron microscopy image of the material, although this is not mandatory if the technique is not available to the authors.

In summary, I believe the manuscript could be published after these minor changes.

Comments on the Quality of English Language

The article requires some improvements in English. There are no glaring grammatical errors, but I believe an English revision would enhance the text and make it more comprehensible.

Reviewer 4 Report

Comments and Suggestions for Authors

Manuscript ID: processes-28386407 " Preparation of high specific surface area activated carbon from petroleum coke by KOH activation in a rotary kiln" requires several additions before it can be accepted for publication.

Below, you will find detailed comments:

(1) In section 2.3.2, Characterizations of AC, please state the degassing conditions of the test samples.

(2) Please section 2.3.2 Characterizations of AC to verify that actually p/po=0.98 and not 0.96 was used to determine the texural paramters?

(3) Please discuss the results for SBET, Da, Vt, Vmicro, found in Table 3, as the results for SBET, Da are found in the abstracts and summary.

(4) The authors did not present the results concerning macropores, but they refer to this parameter in various parts of the article. On the basis of the research carried out, presentation of Vmezo is possible.

(5) Section 4 should be modified to the most important results.

The paper presents important results in the industrial preparation of activated carbons and can be accepted for publication after addressing the mentioned comments.

Round 2

Reviewer 1 Report

Comments and Suggestions for Authors

Super! Thank you!

Reviewer 2 Report

Comments and Suggestions for Authors

The authors took into account all the reviewer's comments. Thank you. The text of the manuscript and the originality of the work have become clearer.